# The Use of Collagen Matrix in the Treatment of Gingival Recession—A Pilot Study

**DOI:** 10.3390/jpm12111902

**Published:** 2022-11-15

**Authors:** Marlena Pedowska, Marta Prokop, Renata Chałas, Maja Ptasiewicz

**Affiliations:** 1Department of Oral Medicine, Medical University of Lublin, 20-093 Lublin, Poland; 2Mazovian Center of Dentistry, 00-301 Warsaw, Poland

**Keywords:** gingival recession, collagen matrix, keratinized gingivae, coronally advanced flap, root coverage

## Abstract

Background. Gingival recessions are common mucogingival deformities and conditions around teeth, which are described as the apical migration of tissues in the relation to the cementoenamel junction. One of the types of graft materials used to treat these recessions is the collagen graft material. The aim of this pilot study was to evaluate the effectiveness of the use of the collagen matrix combined with the coronally advanced flap surgery method to cover dental recessions, after a 12 month follow-up period. Methods. The following parameters were assessed in 20 patients, with 38 single or multiple Miller class I or II gingival recessions: depth of the recession—RD; width of the recession—RW; height of the keratinized tissue—HKT; and thickness of the keratinized gingivae—GT. The percentage of the root coverage after 12 months was also calculated. Results. There was a significant difference in the RD and RW values before and after the procedure. On average, the RD decreased by 3.39 mm, and the RW by 3.87 mm. Moreover, the values of the GT and HKT, before and after the treatment, significantly increased by 0.98 mm and 1.13 mm, respectively. The partial root coverage was 43.48%, and the total root coverage was 56.52%. Conclusion. The use of the collagen matrix with a split thickness gingival flap to cover the gingival recessions is a predictable and effective method when the clinician carefully follows the recommendation of the producer, which was confirmed in a 1 year follow-up with good clinical results.

## 1. Introduction

Gingival recessions are common mucogingival deformities and conditions around teeth, which are described as the apical migration of tissues in the relation to the cementoenamel junction (CEJ). Their etiology is multifactorial. Exposed dental roots frequently coexist with dentin and enamel defects after very intensive toothbrushing, with intrasulcular cervical restorations or with periodontal disease. Gingival recessions are more commonly seen among patients with the thin periodontal phenotype, as well as among patients with buccal inclination or rotated teeth; especially where teeth are positioned into the vestibular direction. Orthodontic treatment may also have a negative impact on the gingival margin. Movement of teeth in the buccal or labial direction, especially among patients with the thin periodontal phenotype, can lead to the development of gingival tissue recession. Hence, nowadays, the clinical and radiological evaluation of the phenotype of gingival and bone before orthodontic correction is essential. Clinical observations lead to a conclusion that the thick phenotype plays a protective role for the development of the recession, especially during a treatment where labial or buccal movements are performed [1].

The treatment of gingival recession often depends on the patient’s dental awareness and aesthetic needs. The majority of adult patients have at least one gingival recession, but not all gingival recessions need to be surgically treated. When it is necessary, and the patient is qualified for a surgical treatment after clinical examination, several dozen techniques and flap designs are used to correct the gingival recession. Procedures using pedicle flaps, free soft tissue grafts, a combination of pedicle flaps, and autogenous subepithelial connective tissue grafts, as well as enamel matrix derivate and membranes are widely described as being effective [2]. Some studies also show the efficacy of the coronally advanced flap technique (CAF) in covering recessions, especially in a patient with aesthetic requests and if there is adequate keratinized tissue apical or lateral to the recession. In these surgical approaches, the soft tissue utilized to cover root exposure is similar to that originally present, and thus the aesthetic result is satisfactory. Furthermore, the post-operative discomfort is minimal as second surgical sites (palate) are not involved. However, for deeper and larger recessions, especially coexisting with the thin tissue phenotype, a coronally advanced flap seems to be less effective in comparison when it is used together with autogenous grafts or another alternative material [3]. Another option is covering the exposed roots with autogenous tissue, but for that we need to harvest it from the palate. Not every palate offers adequate connective tissue graft thickness and size. The most important problem in obtaining a palatal transplant is the risk of a damage to the palatine artery; therefore, a very careful preoperative assessment of the maximum size of the palatal tissue is required. For patients, it means one more incision and another wound to heal, which may cause a discomfort or pain. Every palate offers a different quality (thickness) or quantity (size) of the connective tissue, and sometimes they inadequate for the size of the recessions. Those limitations prompted researchers to seek for alternatives for connective tissue grafts [4]. One of the types of graft materials that are used in periodontal surgery is the collagen graft material. This collagen matrix (CM) is designed to initiate soft tissue regeneration and provide suture stability with immediate support for the blood clot and the early colonization of soft tissue cells (blood cells and nerve cells) [5]. The dense outer layer, which consists of collagen fibers with cell-protective properties, not only protects against bacterial invasion, but also has a certain elasticity that simplifies the suturing process. The second layer, consisting of a porous, dense and spongy structure of collagen, allows an easy formation of a coagulum and promotes tissue integration and angiogenesis [6]. The representative of the collagen matrix—Mucograft^®^ (Geistlich Pharma) is one of the alternative options, primarily on account of its structure, which is composed of collagen types I and III and its three-dimensional form [7].

The aim of this pilot study was to evaluate the effectiveness of the use of the collagen matrix combined with the coronally advanced flap (CAF) surgery method in covering dental recessions, after a 12 month follow-up period. The following parameters were assessed: depth of the recession—RD; width of the recession—RW; height of the keratinized tissue—HKT; and thickness of the keratinized gingivae—GT. The percentage of the root coverage, thickness of soft tissue over the root, recession width, and keratinized tissue width, after 12 months, were also calculated.

## 2. Materials and Methods

### 2.1. Design of the Study and Patients’ Qualification

The study was designed as a single center, performed by the operator, using the same techniques regarding single or multiple type recessions. The assessment of the clinical oral parameters was performed by one experienced dentist (with 3 years of experience in periodontal surgery) during the first as well as control examinations, with a basic dental diagnostic kit and periodontal probe. The reproducibility of the operator was calibrated prior to the start of the project. Five patients with class I and II recessions, not included in the study, were examined with a periodontal probe CP-15 (Orimed, Osieck, Poland). The measurements were made twice one week apart. The clinical parameters (RD, RW, HKT, GT) were assessed in the same way as in the main study. The calibration was accepted when the base and after-one-week measurements did not differ by more than 0.5 mm in 90% of the measurements. The study design was accepted before its execution by the Bioethics Committee of the Medical University in Lublin, Poland (Agreement Nr. 0254/312) and performed at the Department of Oral Medicine.

#### 2.1.1. Eligibility Criteria

All 20 patients were selected according to the following principles:

Inclusion criteria:Presence of at least one localized or multiple gingival recessions, according to Miller’s classification—class I and II;Cementoenamel junction was visible in the teeth qualified for the root coverage procedure;All patients demonstrated adequate plaque control, the FMBOP (full-mouth bleeding on probing score) according to Ainamo and Bay [8] was below 15%, and the API (approximal plaque index) according to Lange et al. [9] was less than 25%;All patients were at least 18 years old.

Exclusion criteria:
Smoking;Pregnancy;Allergy to collagen;Diabetes or any systemic disease which may affect the healing process of the oral mucosa;Taking steroids.

Each participant of this study signed an informed consent form in accordance with the Helsinki Declaration of 1975 as revised in 2000.

The total number of examined patients was 20, and 38 recessions were covered. All participants had single or multiple Miller class I or II gingival recessions. After a year, 6 patients did not show up for the control check-up. The final analysis of the gathered data was based on 14 patients and 23 gingival recessions.

#### 2.1.2. Clinical Measurements

After the qualification of the patients, before the procedure, and 12 months after the procedure, the following parameters were assessed (in millimeters with the use of a periodontal probe): depth of the recession—RD; width of the recession at the cementoenamel junction—RW; height of the keratinized tissue (from the gingival margin to the mucogingival junction)—HKT; and thickness of the keratinized gingivae with the injection needle and the silicone marker, 1 mm above the gingival margin—GT. The aesthetic effect was also assessed 12 months after the procedure. The diagnostic instrument was a CP-15 periodontal probe with laser calibration every 1 mm. Pre-operative and follow-up photographs were taken at each visit.

#### 2.1.3. Treatment

All participants of the study had a full mouth periodontal examination with a registration of the periodontal parameters. Once the selected participants agreed to participate in the study, they were provided with customized oral hygiene instructions, including control of traumatic tooth brushing techniques as well as dental prophylaxis and polishing. Particular attention was paid to the elimination of potential risk factors for recession. The occlusal interferences in the centric and eccentric occlusion were corrected, and hygiene procedures were performed, including oral hygiene instruction with the method of tooth brushing adjusted to the patient’s needs.

#### 2.1.4. Surgical Procedures

The procedure was performed using a local anesthetic containing 4% articaine with adrenalin (Molteni Dental, Milan, Italy). The coronally advanced technique, according to Zuchelli and De Sanctis, split thickness flap was utilized in the study. The choice of surgical method was dictated by recommendations of the producer (Geistlich Pharma, Wolhusen, Switzerland) (Figure 1).

The exposed root surface was mechanically cleaned and prepared for the procedure. After the preparation, the collagen matrix Mucograft^®^ (Geistlich Pharma) was cut to the size of the defect, allowing a slight overlap, and placed on the recipient bed with the smooth, more glossy layer facing the prepared flap and the spongy surface facing the periosteum (Figure 2). The surgical technique used in this treatment was the coronary displacement of the plate.

The preparation of the flap size, which completely covers the matrix, is an essential stage of the procedure. Thanks to the three-dimensional structure, this matrix ensures the growth of tissue. However, for this process to occur, it must have contact with vascularized tissue on each side. Moistening the collagen matrix with blood already gives some stability, but according to the recommendation of producer, the material was fixed to the surface by resorbable sling sutures (6-0). While placing the suture pressure on the matrix was avoided, only the gentle assistance of the pliers was allowed. When the collagen matrix is soaked by blood, it is more prone to tear.

Subsequently, the exposed root and collagen matrix was covered with the coronally advanced flap and stabilized with single sutures (non-resorbable 6-0) and sling suture. No periodontal dressing was applied to the wound (Figure 3).

### 2.2. Post-Surgical Protocol

After the surgery, each participant of the research was instructed to: rinse twice a day with chlorhexidine mouthwash (0.2%) and apply chlorhexidine gel once a day for 4 weeks; follow a soft diet and avoid any mechanical trauma; avoid tooth brushing of the surgical area for 3 weeks; reduce the post-operative oedema with a recommended cold compress—gentle and without pressure; use prescribed pain killers and anti-inflammatory drugs, according to the individual needs (e.g., ibuprofen 400 mg). Sutures were removed after 14 days, patients were recalled after 12 months post-operative, and photographs were taken on the control visit.

#### 2.2.1. Data Analysis

The main hypothesis was that CAF, combined with the xenogenic collagen matrix, would be able to improve the clinical parameters by increasing the keratinized gingivae height and thickness, as well as minimizing or totally covering the existing recession.

#### 2.2.2. Statistical Analysis

The obtained results were analyzed statistically using Statistica 9.1 computer software (StatSoft, Kraków, Poland). The values of the analyzed quantitative variables were presented as mean, median, lower and upper quartile values, minimum and maximum values and standard deviation, and qualitative variables by means of the count and percentage.

The Chi^2^ test, with the Yates’s correction, was used to check the relationship between the occurrence of total recession coverage and the location of the teeth. The normality of the distribution of variables was assessed with the Shapiro–Wilk test. The Wilcoxon pairwise test was used to test the differences between the “before” and “after” measurements. A significant level of *p* < 0.05 was adopted, indicating the existence of statistically significant differences or relationships.

## 3. Results

Each patient was treated for single or multiple recessions. In eight patients, the procedure was performed on one tooth, in five patients, it was performed on two teeth, and in one patient, five recessions were covered (Table 1).

In nine cases the recession occurred at the incisors (39.13%), in eight at the canines (34.78%), and in four (17.39%) at the premolars. The procedure was performed in two cases at the molars (8.70%) (Table 2).

The total root coverage 1 year after the surgical treatment was in 13 recessions (56.52%), and the partial root coverage was in 10 cases (43.48%) (Table 3).

The partial root coverage was found in seven anterior and in three posterior teeth. There was no statistically significant correlation between the occurrence of total root coverage and the location of the anterior–posterior teeth (Table 4).

There was a statistically significant difference (*p* < 0.001) in the RD and RW values before and after the procedure. On average, the RD decreased by 3.39 mm and RW by 3.87 mm. Additionally, the values of the GT and HKT before and after the treatment changed statistically significantly and increased by 0.98 mm and 1.13 mm, respectively. The percentage increase was 52.17% for GT and 62.32% for HKT. The minimum increase was 0% (no change had taken place), and the maximum increase was 300% (Table 5). The color and consistency of the gingivae did not differ from the surrounding tissue in a clinical observation, and no hypertrophy or scars were observed of 12 months after the surgical treatment (Figure 4).

Twelve months after the procedure, the expert periodontist subjectively assessed the achieved aesthetic result. Color matching, consistency, and surface texture of the formed tissue were taken into account. The final size of the generated keratinized tissue in relation to the initial dimensions was used to evaluate the success.

## 4. Discussion

A realistic assessment of the planned treatment results is one of the most important criteria for making a decision about surgical treatment. The analysis of the patient’s medical history, local factors related to gingival recession, and the factors related to the surgery lead the surgeon to making the decision related to the treatment. The first group of factors includes systemic diseases, e.g., diabetes, which influence the wound healing after surgery. In this group, we also include cigarette smoking, which has a significant impact on the prognosis of surgery. The second group includes local conditions with no visible signs of inflammation [10] and oral hygiene defined by the API index. Only optimal local conditions allow for precise and successful surgery. The final group of factors essential for therapeutic success are the conditions of the surgical procedure itself: flap thickness determined by the gingival phenotype, thickness of the connective tissue graft in procedures with the use of autogenous connective tissue, which should not exceed 1.5 mm, and excessive correction of the procedure [11]. All of this criteria were fulfilled in our study, which can explain the high percentage of recession covering—85%.

Gingival recession development (GR development) has been associated with the exposure of the root surface in the mouth, due to the displacement of the gingival margin apical to the cementoenamel junction, but also with functional impairment, aesthetic problems, and many other dental-related conditions. The parameters concerning the condition of the periodontal tissues, in the interdental spaces adjacent to the recession and the size of the defect itself, are essential in the prognosis of the coverage of gingival recession. Full coverage of the exposed root surface can be expected in Miller class I and II recessions, without bone loss in the interdental spaces [4]. Therefore, in our own study, the authors qualified only patients with class I and II recession, according to Miller. Relatively small gingival recessions, not exceeding 5 mm in height and 3 mm in width, seem to have the best chance of a complete surgical coverage. The post-operative stabilization of the flap should be 1–1.5 mm coronally to the expected final gingival margin level. The shape of the flap should ensure a proper blood supply, and the stress-free suturing technique plays a most important role in the surgical procedure. Proper post-trial management and patient compliance are also important [5].

A variety of techniques can be used to surgically cover gingival recessions: free epithelial-connective tissue and connective tissue grafts, pedunculated flaps, combination methods, and guided tissue regeneration. In most cases, the treatment of exposed roots demands the harvesting of connective tissue from the palate. Free gingival grafting is the first surgical technique used in periodontal surgery, in which the graft is taken from the palate or from the maxillary tuber. Obtaining a transplant from the area of the maxillary tuber results from the fact that the operating site is small. Healing of the donor site is much simpler and faster than in the case of picking a graft from the palate.

The success of periodontal surgery for the correction of GR defects relies primarily on two parameters: a soft tissue coverage of the exposed root and tissue thickness. The coverage of the denuded root provides the desired aesthetic outcomes, and the increase in tissue thickness ensures long-term stability and prevents further recession.

The collagen matrix Mucograft^®^ is one of the alternative options, primarily on account of its structure, which is composed of collagen types I and III and its three-dimensional form. The effectiveness of recession coverage, described by McGuire and Scheyer, was that 6 months after surgery 83.5% of cases were covered, in comparison with 97% for a subepithelial connective tissue graft [3]. In the 12 month follow-up, the collagen matrix increased to 88.5% and the CTG (guided tissue regeneration) to 99.3%, respectively. As observed by Cardaropoli et al. in a 12 month study, total coverage was achieved for 8 of the 11 recessions, which gives a result of 72%, and the mean root coverage was 94.32% (a mean reduction of 2.86 mm in the height of the recession) [12]. These results do not greatly differ from those in the present study. In the authors’ own study, the average percentage coverage of the root surface of the teeth, 1 year after the procedure, was 85% (14 recessions totally covered in 23 operated). The total root coverage 1 year after the surgical treatment was 60.86%.

According to Cardaropoli et al. [12], a 1.23 mm increase in the width/height of the keratinized gingivae and a 1 mm increase in the thickness of the gingival were observed. In the research by Dominiak et al. [13], an increase in HKT (height keratinized tissue) was obtained after 6 months at a level of 1.44 mm, and after a further 6 months, by a further 0.34 mm. In the author’s own study, the average increase of the gingival height was 2.3 mm, whereas the average increase of the gingival thickness was 1 mm. Such a significant increase in this parameter is very important, especially due to the search for a method that allows obtaining favorable results in long-term evaluation.

When assessing post-surgical aesthetics, the results of both the authors own observations and those of McGuire and Scheyer were comparable [3,11]. The authors used the visual assessment of the color and structure of tissues after the procedure. Clinical evaluation was performed independently of the operator. Dominiak et al. [13] assessed post-treatment aesthetic effects using the Bouchard Classification, which defines post-treatment aesthetics on a multi-stage scale. The system devised by Bouchard and co-authors is commonly used in assessment of post-surgical aesthetics. According to the obtained results, in most cases, the aesthetics were better than before the procedure, and there were no differences between the treatment area and the surrounding tissues. Similar results were obtained for the method of covering the recession with a collagen membrane or for tissue culture [14,15,16]. In all cases, the author of the research was content with the color and shape of the gingivae. No color or slight color difference was noticed in comparison to the adjacent tissues [17,18].

Górski et al. [19], in their study, compared the outcomes of the modified coronally advanced tunnel technique (MCAT), combined with subepithelial connective tissue graft (SCTG) with or without enamel matrix derivative (EMD), in the treatment of gingival recession types 1 and 2. Both treatments showed high aesthetic results. The root coverage aesthetic score in the SCTG+EMD group was 9.2 ± 1.2, whereas in the SCTG group it was 8.7 ± 1.3. In addition to the average root coverage esthetic score (RES), there were also statistically significant differences in the three different parameters: marginal tissue contour, muco-gingival junction alignment, and gingival color, between two treatment modalities. All of the above mentioned were in favor of test sides, and keloid formation was not observed in any patient after 6 months [19].

Due to the use of the collagen matrix, it is possible to avoid the excessive gingival thickness, which is sometimes observed in the surgical methods with connective tissue grafts [20,21]. The good aesthetics of procedures with Mucograft^®^ is connected to the high porosity of the material compared to the different collagen membranes, which is 90%. The porous layer is thicker and induces the growth of soft tissues into it.

The limitations associated with the collagen matrix during the procedure, compared to an autogenous transplant, included its stretchability and lower membrane formation. During the procedure, the collagen matrix was easier to damage than the autogenous graft of subepithelial connective tissue.

CM appears to be a suitable substitute for CAF. It has advantages, such as the increase of keratinized tissue, less pain, shorter procedure times, and proper color matching. Nevertheless, there is a lack of data on the long-term stability of the collagen matrix results, and future studies with long-term follow-ups are suggested [22,23,24,25,26].

## 5. Conclusions

The use of the collagen matrix with the split thickness gingival flap to cover the gingival recessions is a predictable and effective method when the clinician carefully follows the recommendation of the producer what was confirmed in a 1 year follow-up with good clinical results. In all patients, the healing process was shorter and more comfortable, as only the recipient site has to be prepared.

### Limitation of the Study

The following pilot research was carried out on a statistically irrelevant sample. The limitation is a result of the patients’ absence on control visits (six patients missed the follow-up visits). Further observation is in progress.

## Figures and Tables

**Figure 1 jpm-12-01902-f001:**
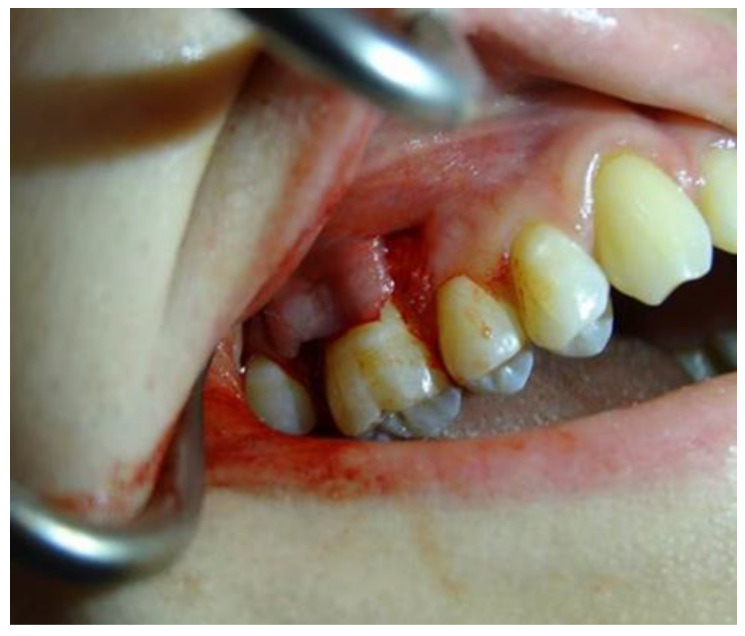
Split thickness flap preparation.

**Figure 2 jpm-12-01902-f002:**
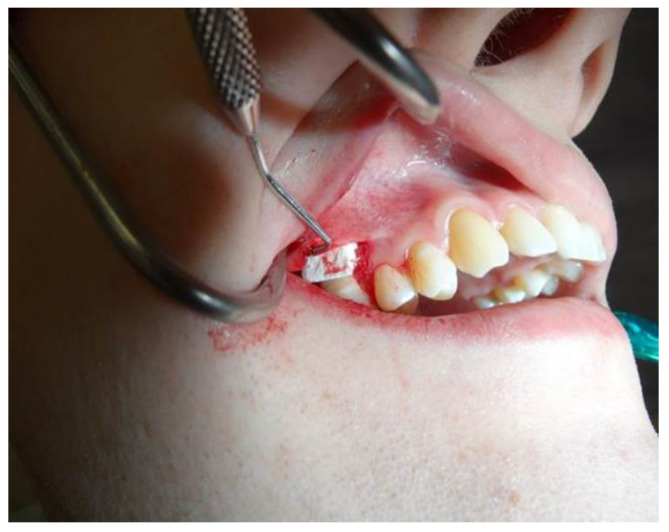
Collagen matrix material placement.

**Figure 3 jpm-12-01902-f003:**
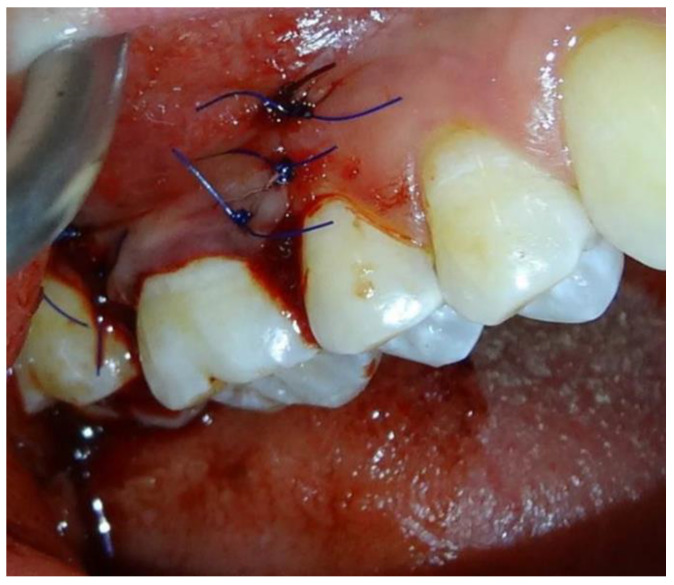
Simple loop suture technique: Flap stabilization with singles sutures and sling sutures.

**Figure 4 jpm-12-01902-f004:**
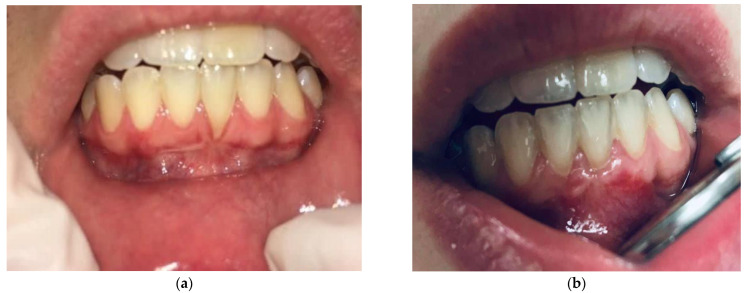
Miller class II recession-tooth 31; (**a**) before surgery, (**b**) after surgery.

**Table 1 jpm-12-01902-t001:** The values of periodontal parameters (mm) before and after the procedure.

Patient	Tooth Number	RD Before	RD After	RW Before	RW After	HKT Before	HKT After	GT Before	GT After
1	24	6	3	5	4	2	4	2	3
	25	5	2	5	4	4	6	2	3
2	24	4	0	5	0	4	5	2	3
3	21	4	0	5	0	4	4	2	3
	22	4	0	4	0	5	5	2	3
	11	3	0	6	0	2	4	2	3
	12	3	0	4	0	3	4	2	3
	13	3	0	4	0	4	4	2	3
4	16	4	0	8	0	4	4	1.5	3
	26	4	0	4	0	3	4	1.5	3
5	31	5	0	3	0	1	3	1.5	3
6	13	4	0	7	0	4	5	2	3
	14	4	2	5	4	1	4	2	3
7	23	6	0	10	0	3	4	2	3
8	23	4	1	4	3	6	6	2	3
	11	5	2	9	4	3	4	2	3
9	21	4	1	9	4	2	3	2	3
	42	4	0	3	2	3	4	2	2
10	43	4	1	5	2	3	4	2	2
11	13	3	0	4	0	6	7	2	3
12	23	3	0	4	0	6	7	2	3
13	12	4	2	3	3	2	3	2	3
14	13	3	1	6	3	1	4	2	3

Depth of the recession—RD; width of the recession—RW; height of the keratinized tissue—HKT; thickness of the keratinized gingivae—GT.

**Table 2 jpm-12-01902-t002:** Recessions location in examined patients.

		N	%
Location	I quadrant	10	43.48
	II quadrant	10	43.48
	III quadrant	1	4.35
	IV quadrant	2	8.69
Tooth number	1	5	21.74
	2	4	17.39
	3	8	34.78
	4	3	13.04
	5	1	4.35
	6	2	8.70
	maxilla	20	86.96
	mandible	3	13.04
	anterior teeth	17	73.91
	posterior teeth	6	26.09
	incisors	9	39.13
	canines	8	34.78
	premolars	4	17.39
	molars	2	8.70

**Table 3 jpm-12-01902-t003:** Root coverage after 1 year.

		N	%
Root coverage	total	13	56.52
	partial	10	43.48

The total root coverage was obtained in anterior teeth in 10 cases, and in 3 cases in posterior teeth.

**Table 4 jpm-12-01902-t004:** Dependence of the total root coverage and anterior–posterior tooth location.

Localization	Total Root Coverage	Partial Root Coverage	
anterior teeth	10	7	Chi^2^_Yate’s_ = 0.011 *p* = 0.917
	58.82%	41.18%	
posterior teeth	3	3	
	50.00%	50.00%	
total	13	10	

**Table 5 jpm-12-01902-t005:** Values of the measured parameters before and after the procedure.

Value (mm)	M	Me	Min	Max	Q1	Q3	SD	
RD before	4.04	4	3	6	3	4	0.88	Z = 4.197 *p* < 0.001
RD after	0.65	0	0	3	0	1	0.93	
RD difference before and after	3.39	3	2	6	3	4	0.94	–
RW before	5.30	5	3	10	4	6	2.01	Z = 4.107 *p* < 0.001
RW after	1.43	0	0	4	0	3	1.75	
RW difference before and after	3.87	4	0	10	1	5	2.46	–
HKT before	3.30	3	1	6	2	4	1.52	Z = 3.724 *p* < 0.001
HKT after	4.43	4	3	7	4	5	1.12	
HKT difference before and after	1.13	1	0	3	1	2	0.87	–
HKT % difference before and after	62.32	33.33	0	300	16.67	50	87.03	–
GT before	1.93	2	1.5	2	2	2	0.17	Z = 4.015 *p* < 0.001
GT after	2.91	3	2	3	3	3	0.29	
GT difference before and after	0.98	1	0	1.5	1	1	0.35	–
GT % difference before and after	52.17	50	0	100	50	50	23.73	–

M—average, Me—median, Min—minimum value, Max—maximum value, Q1—lower quartile, Q3—upper quartile, SD—standard deviation, Z—result of the Wilcoxon order of pairs test, *p*—statistical significance.

## Data Availability

The data presented in this study are available on request from the authors.

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
