# Peer review of "The Use of Collagen Matrix in the Treatment of Gingival Recession—A Pilot Study"

_jpm, 2022, doi:10.3390/jpm12111902_

Round 1

Reviewer 1 Report (New Reviewer)

More samples and double blind design were needed in the future.

Data of almost 1/3 patients were not gatherd at the end of this study.

Only one operator was mentioned in this study.

Author Response

Thank you for  important comment. In the future studies, we plan a larger study group and double blind design.

The study was designed as a single center, performed by the operator, using the same techniques regarding single or multiple type recessions. The assessment of clinical oral parameters was performed by one experienced dentist (3 years of experience in periodontal surgery) during first as well as control examinations. Reproducibility of the operator was calibrated prior to the start of the project.

Reviewer 2 Report (New Reviewer)

This study investigated the usefulness of a collagen matrix for repairing gingival recession through a human clinical experiment. This is a meaningful topic for clinicians who want to repair the gingival recession. 

The authors made thorough evaluations in clinics and objectively demonstrate the results. The conclusion is appropriate to be used by other clinicians or researchers. 

I recommend improving the quality of the photos. 

I do not see any other red flags in the manuscript. 

Author Response

Thank you for your positive review. We tried to improve the quality of the photos.

Reviewer 3 Report (New Reviewer)

This is a correctly written document with great clinical utility, however the following changes are suggested: 

*Perhaps in the title indicate pilot test as indicative, because the sample is small.

*In introduction, highlight the justification for this technique. 

*Mention that the research project was approved by an ethics committee before its execution, if possible provide the code. 

Author Response

This is a correctly written document with great clinical utility, however the following changes are suggested: 

*Perhaps in the title indicate pilot test as indicative, because the sample is small.

Thank you for your review and relevant comments. We have added information about the pilot study in the title of the manuscript.

*In introduction, highlight the justification for this technique. 

In introduction, we have highlighted the justification for CAF technique. 

*Mention that the research project was approved by an ethics committee before its execution, if possible provide the code. 

Information about the consent of the Bioethics Committee was in the text- line 96-97.

We have corrected the information on the Bioethics Commission's consent.

“The study design was accepted before its execution by the Bioethics Committee of the Medical University in Lublin, Poland (Agreement Nr. 0254/312) and performed at the Department of Oral Medicine.”

Reviewer 4 Report (New Reviewer)

This is a clinical research study using collagen matrix, which is useful because it includes a statistical study, but the selection and exclusion criteria for patients are unclear. Therefore, we consider that this study is not clinical research and can only be accepted as a Case Series.

Table2frontanterior

              back→posterior

Author Response

Thank you for your review.  

We have added the paragraph 2.1.1 Eligibility Criteria

All 20 patients were selected according to the following criteria:

Inclusion criteria:

-Presence of at least one localized or multiple gingival recession according to Miller’s classification –Class I and II;

-Cemento-enamel junction was visible in the teeth qualified for the root coverage procedure;

- All patients demonstrated adequate plaque control, the FMBOP (full‑mouth bleeding on probing score) according to Ainamo and Bay [8] was below 15%, and API(approximal plaque index) according to Lange et al. [9] was less than 25%;

-All patients were at least 18 years old.

Exclusion criteria:

-Smoking;

-Pregnancy;

-Allergy to collagen;

-Diabetes or any systemic disease which may affect the healing process of the oral mucosa;

- Taking steroids

We have  changed front to anterior and back to posterior in Table 2.

Round 2

Reviewer 1 Report (New Reviewer)

1."Anterior teeth/front teeth, posterior teeth/back teeth" could be seen in one page. Which one was correct presentation?

2. Was there any differnce between  the results of anterior teeth and posterior teeth. And plesease explain the resean why did it happened?

Author Response

1."Anterior teeth/front teeth, posterior teeth/back teeth" could be seen in one page. Which one was correct presentation?

We changed the front  teeth to anterior and back teeth to posterior according to the reviewer's opinion, the final version is anterior and posterior. The change is  also introduced in text now.

  1. Was there any differnce between  the results of anterior teeth and posterior teeth. And plesease explain the resean why did it happened.

There was no statistically significant correlation between the occurrence of total root coverage and the location of the anterior-posterior teeth (Table 4).

Inappropriate tooth brushing and dental plaque buildup may lead to early gingival recession. Decreased alveolar bone crest thickness, combined with delicate gingival margin, commonly found in maxillary canines and mandibular incisors may also be the reason of gingival recession. With dehiscence and fenestration, the chances of gingival recession occurring are much higher. Those bone crest morphological defects are predisposing factors of gingival recession are more frequently found in teeth malpositioned in the dental arch, especially uprighted teeth, as it is the case of canines subjected to orthodontic traction and which had erupted more uprightedly. Labial and lingual  movements of frenulas  inserted near the cervical region of gingiva as well as cicatricial adhesions might predispose the region to gingival retraction.

Reviewer 4 Report (New Reviewer)

I think it has been properly corrected and is fine.

Author Response

Thank you for your positive opinion.

This manuscript is a resubmission of an earlier submission. The following is a list of the peer review reports and author responses from that submission.

Round 1

Reviewer 1 Report

All comments have been addressed

Author Response

Thank you for the review.

Reviewer 2 Report

Very few of the suggestions raised by the reviewer have been solved. Please find a few examples bellow.

The authors ignored all the suggestions of the reviewer concerning the abstract. Yes, in the main text there were mentions about the surgical technique, statistical analysis etc…. but not in the abstract! Thus, the information provided in the abstract does not allow the redear to imagine the development of the research, nor its aim and outcomes. Moreover, it is not normal to provide information in abstract introduction to the detriment of the explanations in the M&M.  

Consequently to the request of the reviewer, the authors modified the aim of the study. However, we do not know what exactly the authors mean with the term “coronary displacement of the plate surgery method”. What plate? Probably they refer to coronally advanced flap. The results provide more data than that proposed to be obtained in the aim of the study. None of the other outcomes provided in the “Results” section are mentioned in the aim of the study. Also, other outcomes mentioned in M& M/statistical analysis were not considered in the aim of the study.

The authors mentioned they did not understand the reviewer observations that the study outcomes were not clearly mentioned. The consecrated outcomes provided after muco-gingival interventions are: the success rate, predictability and keratinized tissue dimensions. No interest for the reader the descriptive analysis concerning type of the teeth etc.

And yes, even is one examiner, the intra-examiner reproducibility must be calculated!

The reviewer previously mentioned: “WHO probe is not suitable to monitor GR outcomes. This is a major concern because the used instrument does not ensure a precise, reproducible evaluation. However, no manufacturer name nor its coordinates are provided. And the authors answered: “Thank you for your precision. According to the comment this information has been corrected.” BUT NO, THE AUTHORS DID NOT CORRECTED THIS ISSUE!!! WHO PROBE WAS CHANGED WITH “BASIC DENTAL EXAMINATION KIT”. THIS IS NOT ENOUGH SPECIFIC AND THIS AMBIGOUS MENTION OF THE EXAMINATION TOOL COULD NOT SUSTAIN THE VERIDICITY OF THE METODOLOGY.

I do not question the expertise of the statistician only that of the other authors who seem to be unable to understand the observations made by the reviewer (for example regarding the outcomes). However, the major problem remains the methodology that I consider unreproducible.

Author Response

Thank you for all comments and suggestions. Comments and questions have been considered and modifications have been done in the revised manuscript. Hereafter, you will find the answer to the questions and comments. We hope that changes suit with recommendations.

Round 2

Reviewer 2 Report

Due to profound methodological lacks, I maintain my previous opinion.